# VL-Rethinker: Incentivizing Self-Reflection of Vision-Language Models with Reinforcement Learning

**Haozhe Wang**[◇♡]**, Chao Qu**[‡]**, Zuming Huang**[‡]**, Wei Chu**[‡]**, Fangzhen Lin**[◇]**, Wenhu Chen**[♡†]

HKUST[◇], University of Waterloo[♡], INF[‡], Vector Institute[†]

Corresponding to: jasper.whz@outlook.com, wenhuchen@uwaterloo.ca

Project Page: `https://tiger-ai-lab.github.io/VL-Rethinker/`

**Performance Comparisons on Multimodal Benchmarks**

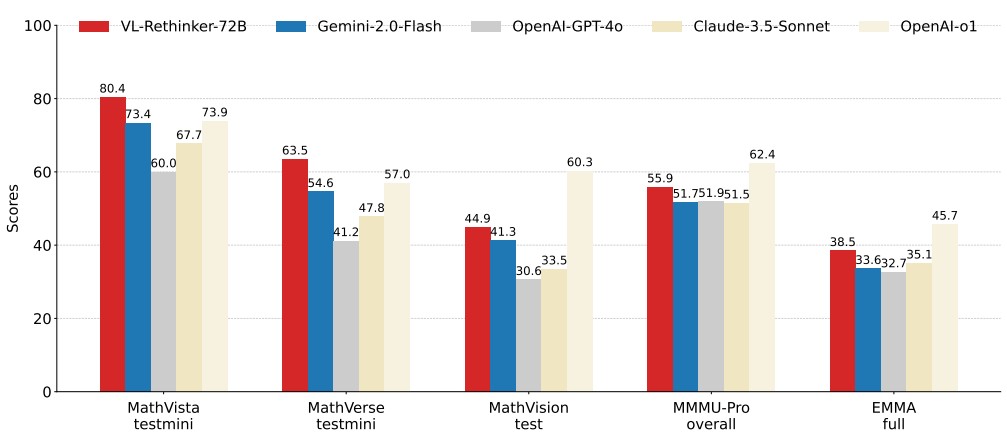

Figure 1: Performance comparison between `VL-Rethinker` and other SoTA models on different multimodal reasoning benchmarks.

## Abstract

Recently, slow-thinking systems like GPT-o1 and DeepSeek-R1 have demonstrated great potential in solving challenging problems through explicit reflection. They significantly outperform the best fast-thinking models, such as GPT-4o, on various math and science benchmarks. However, their multimodal reasoning capabilities remain on par with fast-thinking models. For instance, GPT-o1's performance on benchmarks like MathVista, MathVerse, and MathVision is similar to fast-thinking models. In this paper, we showcase how to enhance the slow-thinking capabilities of vision-language models using reinforcement learning, to advance the state of the art, without relying on costly distillation. First, we adapt the GRPO algorithm with a novel technique called Selective Sample Replay (SSR) to address the vanishing advantages problem. While this approach yields strong performance, the resulting RL-trained models exhibit limited self-reflection. To further encourage slow-thinking, we introduce Forced Rethinking, which appends a rethinking trigger token to the end of rollouts in RL training, explicitly enforcing a self-reflection reasoning step. By combining these two techniques, our model, `VL-Rethinker`, advances state-of-the-art scores on MathVista, MathVerse to achieve 80.4%, 63.5% respectively. `VL-Rethinker` also achieves open-source SoTA on multi-disciplinary benchmarks such as MathVision, MMMU-Pro, EMMA, and MEGA-Bench, narrowing the gap with OpenAI-o1. We conduct comprehensive ablations and analysis to provide insights into the effectiveness of our approach.

39th Conference on Neural Information Processing Systems (NeurIPS 2025).

# 1 Introduction

Recently, slow-thinking systems such as OpenAI-o1 [Jaech et al., 2024], DeepSeek-R1 [Guo et al., 2025], Kimi-1.5 [Team et al., 2025], Gemini-Thinking [Team et al., 2023], and QwQ/QvQ [Bai et al., 2025] have significantly advanced the performance of language models in solving challenging math and science problems. These models engage in extended reasoning and reflection before arriving at a final answer, in contrast to fast-thinking models like GPT-4o [Hurst et al., 2024] and Claude-3.5-Sonnet [Anthropic, 2024], which produce answers rapidly without such deliberation. Through this reflective process, slow-thinking models outperform the best fast-thinking models by over 30% on math datasets such as AIME24 and AMC23 [Hendrycks et al.], and by around 10% on general science benchmarks like GPQA [Rein et al., 2024].

However, their multimodal reasoning capabilities remain on par with fast-thinking models. For example, GPT-o1 achieves 73.9% on MathVista [Lu et al., 2023] and 57.0% on MathVerse [Wang et al., 2024a], which is slightly worse than Qwen2.5-VL-72B [Wang et al., 2024b] scoring 74.8% and 57.2% on the same benchmarks. This raises an important research question:

*How can we effectively incentivize multimodal slow-thinking capabilities in Vision-Language Models?*

To address this, we explore how to effectively train multimodal reasoning models through reinforcement learning (RL), without relying on costly distillation from stronger teacher models [Yang et al., 2025, Deng et al., 2025]. Our main contributions are as follows:

**GRPO with SSR:** We construct a dataset of 38,870 queries covering a diverse range of topics for training our vision-language model (VLM). We adapt the Group Relative Policy Optimization (GRPO) algorithm [Guo et al., 2025], which computes advantages by comparing responses within the same query group and normalizes rewards to guide policy updates. However, we identify a key challenge with GRPO: the vanishing advantages problem. This occurs when all responses in a group receive identical rewards (either all correct or all incorrect), leading to zero advantage signals and ineffective gradient updates. This reward uniformity exacerbates instability as training progresses, hindering the model from exploring deeper reasoning.

To mitigate this, we introduce Selective Sample Replay (SSR), which enhances GRPO by integrating an experience replay mechanism that samples high-value experiences from past iterations. SSR augments the current training batch with rehearsed samples that previously indicated large magnitudes of advantages. This strategic experience replay embodies the principles of curriculum learning [Team et al., 2025] in an online and active fashion Lightman et al. [2023], which dynamically adjusts the training focus towards high-value experiences situated near the model's decision boundaries. While this approach demonstrates strong empirical performance across several multimodal reasoning benchmarks, we observe that the resulting models still exhibit limitations in explicit reflective behavior, suggesting avenues for further improvement.

**Forced Rethinking:** To encourage explicit reflections, we propose a simple yet effective technique called forced rethinking. We append a textual rethinking trigger to the end of roll-out responses and train the model using the same RL setup. This strategy prompts the model to engage in self-reflection and self-verification before producing the final answer. We name the resulting model `VL-Rethinker`. As shown in Fig. 1, `VL-Rethinker` significantly outperforms GPT-o1 on mathematical benchmarks such as MathVista, MathVerse. Furthermore, on general-purpose multimodal benchmarks like EMMA and MMMU-Pro, `VL-Rethinker` achieves a new open-source state of the art performance, closely approaching GPT-o1's performance.

**Observations:** We observe a notable discrepancy between modalities: while RL training often induces slow-thinking behaviors such as longer reasoning traces in math-focused tasks [Zeng et al., 2025, Wen et al., 2025], vision-language tasks rarely exhibit such development. In fact, our analysis reveals that accuracy improvements in VL-Rethinker do not correlate with an increase in the length of the reasoning chain. This finding leads us to a key conjecture: the current bottleneck in multimodal reasoning may not be the *depth* of logical deliberation, but the *accuracy of initial perception*. This suggests the "slow-thinking" incentivized by our method is qualitatively different. Rather than extending abstract logical chains, the model learns to use its rethinking steps to perform perceptual verification and self-correction (e.g., "Wait, let me double-check that number in the chart"). The balance between **perceptual accuracy** over **reasoning depth** – is a critical insight for improving vision-language models and opens a new avenue for future research.

**In summary, our contributions are threefold:** (a) We propose and validate a simple, direct RL approach for enhancing VLM reasoning, offering a viable alternative to complex supervised fine-tuning and distillation pipelines. (b) We introduce Selective Sample Replay (SSR) to mitigate the vanishing advantages in GRPO-based RL for VLMs. (c) We propose Forced Rethinking, a lightweight yet powerful strategy to incentivize self-reflection in VLMs. Our final model, `VL-Rethinker`, sets a new state of the art on key multimodal reasoning benchmarks, demonstrating the value of slow-thinking reinforcement in vision-language modeling.

## 2 Preliminaries

**Problem Formulation** We define the multimodal reasoning task as follows: given a multimodal input consisting of one or more images $I$ and a textual query $Q$, the goal is to generate a textual response $y$ that correctly answers the query by reasoning over both visual and textual information.

Let $\mathcal{V}$ denote the visual input space and $\mathcal{T}$ the textual input space. The input is denoted as $x \in \mathcal{V} \times \mathcal{T}$, where $x = (I, Q)$ captures both modalities. The output is a textual response $y \in \mathcal{Y}$, where $\mathcal{Y}$ represents the response space. The challenge lies in building a vision-language model (VLM) that can integrate multimodal information and perform deep, multi-step reasoning—especially for complex queries requiring extended deliberation or external knowledge.

Our goal is to improve the reasoning capabilities of an instruction-tuned VLM that initially exhibits *fast-thinking* behavior, i.e., producing shallow, immediate responses. We aim to shift the model toward *slow-thinking* behavior [Wang et al., 2025a,b] – engaging in deeper, more deliberate reasoning—to significantly improve performance on downstream multimodal tasks. We achieve this via direct reinforcement learning (RL), which encourages the generation of accurate, thorough, and well-reasoned responses by assigning higher rewards to such outputs.

Formally, we train a policy $\pi_\theta(y|x)$, parameterized by $\theta$, to maximize the expected reward $r(y, x)$ for generating a response $y$ given an input $x$. The reward function $r(y, x)$ is designed to prioritize correctness. The learning objective is:

$$\max_\theta \mathbb{E}_{x \sim \mathcal{D}} \mathbb{E}_{y \sim \pi_\theta(\cdot|x)} [r(y, x)]$$

where $\mathcal{D}$ is a dataset of multimodal queries and their corresponding answers. Consistent with Deepseek R1 Guo et al. [2025], we adopt a binary reward function: $r(y, x) = 1$ if $y$ is correct for input $x$, and $r(y, x) = 0$ otherwise.

**Group Relative Policy Optimization (GRPO)** is a variant of PPO [Schulman et al., 2017]. It estimates the advantages of language model generations by comparing responses within a query-specific group. For a given input $x = (I, Q)$, the behavior policy $\pi_{\theta_{old}}$ generates a group of $G$ candidate responses $\{y_i\}_{i=1}^G$. The advantage for the $i$-th response at time step $t$ is computed by normalizing the rewards across the group:

$$\hat{A}_{i,t} = \frac{r(x, y_i) - \text{mean}(\{r(x, y_1), \dots, r(x, y_G)\})}{\text{std}(\{r(x, y_1), \dots, r(x, y_G)\})}$$

## 3 Our Method

This section outlines our contribution, including Selective Sample Replay (SSR) and Forced rethinking, two techniques to incentivize slow-thinking capabilities.

### 3.1 Vanishing Advantages in GRPO

We identify a critical limitation in GRPO, which we term the "Vanishing Advantages" problem. In GRPO, a simple binary reward signal is used to indicate the correctness of a response $y$ to a given vision-language query $x$. When all responses within a query group are uniformly correct or uniformly incorrect, the calculated advantages become zero for every response in that group. Consequently, such examples cease to provide effective policy gradients, as the gradient signal relies on non-zero advantages to guide learning.

This issue becomes increasingly pronounced as training progresses, especially for high-capacity models. As illustrated in Fig. 2, tracking the training of Qwen2.5-VL-72B reveals a steady decline in the percentage of examples exhibiting non-zero advantages, falling from approximately 40% at the start to below 20% after $16 \times 16$ gradient steps. This decline is a symptom of the policy's tendency

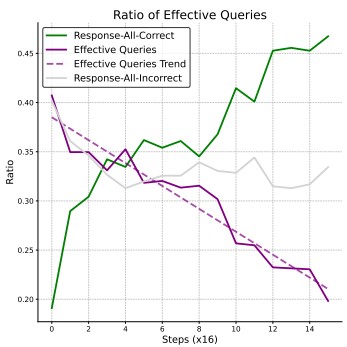

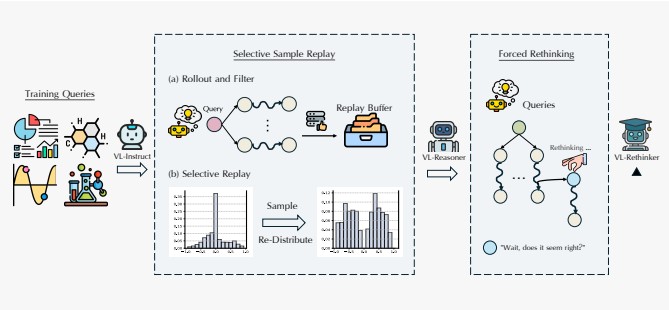

Figure 2: **The Vanishing Advantages problem.** Training of 72B rapidly saturates, leading to a significant decrease of effective queries to only 20% within 256 steps.

Figure 3: **Method Overview.** We present a two-stage RL method based on Qwen2.5-VL-Instruct. The first stage enhances general reasoning through GRPO with Selective Sample Replay (SSR), which retains explored trajectories with non-zero advantages and selectively replay samples based on their advantages. The second stage promotes deliberate reasoning using forced rethinking, where we append a specific rethinking trigger.

to converge towards generating responses that yield uniform rewards within a group over time. As the policy improves and generates more consistently correct and incorrect responses within a query group, the reward diversity (variations) necessary for calculating meaningful advantages diminishes, thereby intensifying the problem. We notice that similar trends have been concurrently observed in GRPO training on text-based LLMs [Yu et al., 2025].

The "Vanishing Advantages" phenomenon undermines the goal of fostering deliberate, complex reasoning in VLMs. As more query groups yield zero advantages, the effective batch size for training shrinks, causing training instability. This instability increases the risk of premature convergence to shallower reasoning traces, discouraging the model from exploring deeper reasoning pathways.

## 3.2 Selective Sample Replay (SSR)

To counteract the Vanishing Advantages problem and maintain training efficiency, we introduce Selective Sample Replay (SSR). SSR enhances GRPO by integrating an experience replay mechanism that strategically samples high-value experiences from past iterations, similar to Prioritized Experience Replay [Schaul et al., 2015] in Temporal Difference learning.

SSR maintains a replay buffer $\mathcal{B}_{\text{replay}}$ that persists for $K$ storing tuples $(x, y_i, \hat{A}_i)$. Critically, the buffer exclusively stores samples for which the corresponding query group exhibited non-zero ($|\hat{A}_k| > 0$). The effective training batch is augmented at each training step by incorporating rehearsal samples drawn from $\mathcal{B}$replay. The sampling is prioritized based on the absolute magnitude of the advantages, thereby emphasizing the rehearsal of experiences that previously indicated significant positive or negative advantage signals. Specifically, a sample $j$ from the buffer is selected with probability:

$$P(\text{select } j) = \frac{|\hat{A}_j|^\alpha}{\sum_{k \in \mathcal{B}_{\text{replay}}} |\hat{A}_k|^\alpha} \tag{1}$$

where $\alpha$ is a hyperparameter that governs the intensity of prioritization. We provide an algorithm diagram for SSR in the appendix.

By selectively sampling valuable experiences, SSR counteracts the issue of vanishing advantages and provides more consistent gradient signals. This stabilizes training and prevents premature stagnation, as further substantiated in the ablation studies (Fig. 5). Furthermore, SSR embodies the principles of curriculum learning [Team et al., 2025, Wang et al., 2022] in an online and active fashion Lightman et al. [2023]. Instead of relying on a static, offline data curriculum, SSR dynamically prioritizes experiences that lie near the model's decision boundaries. This dynamic focus directs training efforts towards improving performance on challenging queries associated with large positive advantages (signaling promising reasoning pathways) and penalizing incorrect solutions corresponding to large negative advantages (often relating trivial queries).

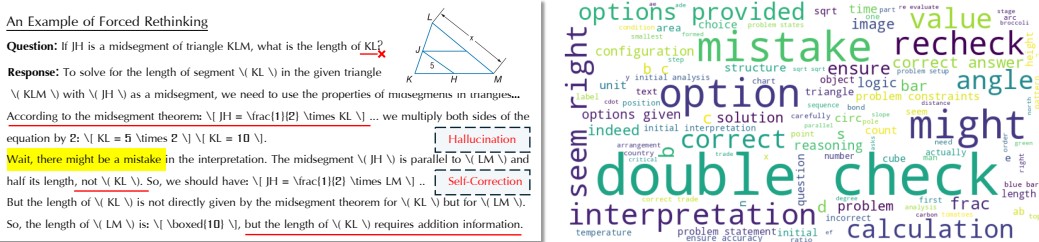

Figure 4: An example of Forced Rethinking (Left). `VL-Rethinker` discovers a flawed problem via rethinking upon its hallucinations. The word cloud of `VL-Rethinker` (Right) shows the learned rethinking pattern of self-verification, self-correction and self-questioning.

## 3.3 Forced Rethinking

While GRPO with SSR improves optimization stability, we observe that *complex, deliberate thinking patterns, such as explicit self-correction, did not consistently emerge as a direct result of standard RL on VLMs*, a divergence from trends observed in large text-only models. Specifically, the base model, Qwen2.5-VL-Instruct, did not intrinsically generate reasoning processes incorporating self-reflection. To explicitly cultivate deliberate reasoning within our VLM framework, we introduce a training technique termed *Forced Rethinking*. This method aims to proactively encourage the model to engage in more extensive internal deliberation before producing a final answer.

Forced Rethinking employs two means to stimulate the model's deliberate reasoning. The first, a straightforward means, involves a hint within the instruction prompt itself, e.g., "regularly perform self-reflection on your ongoing reasoning". This contextual cue serves to increase the model's propensity for generating rethinking sequences. The core principle of Forced Rethinking, however, lies in a targeted intervention within the RL rollout procedure, as depicted in Fig. 3. Following the VLM's initial generation of a response $y_1$ to a given input $x$, we append a specific textual "rethinking trigger" to $y_1$. This augmented sequence is then fed back into the model, urging it to generate a subsequent response segment $y_2$. Consequently, the complete generated sequence becomes $y = y_1 \oplus \text{trigger} \oplus y_2$. To elicit a diverse range of reasoning behaviors, we designed three distinct categories of triggers: self-verification, self-correction, and self-questioning. Detailed descriptions of these rethinking triggers are provided in the appendix.

We apply forced rethinking to a fraction $0 < q < 1$ of the generated responses, and retain only those rethinking trajectories that lead to a correct final answer. Based on these successful forced rethinking trajectories, we incorporate an additional imitation loss (the SFT loss), which directly incentivizes the model to generate the desired deliberate thinking patterns.

Our method shares similarities in forced prompting with *inference-time* budget forcing in S1 [Muennighoff et al., 2025], but it serves as a *training intervention* to incentivize deliberate reasoning. This approach also constitutes a key distinction from methods [Deng et al., 2025, Yang et al., 2025] that rely on SFT distillation from existing deep-thinking systems. Our `VL-Rethinker`, trained with this strategy, does not necessitate a rethinking step for every query. Instead, it learns to strategically engage in this process only when it implicitly determines it to be necessary, potentially leading to more efficient inference. Intriguingly, as illustrated in the example provided in Fig. 4, our `VL-Rethinker` demonstrates the capability to even identify flaws in the given problem when checking its initial reasoning through rethinking, showcasing a form of emergent metacognitive ability (similar to the findings in Wang et al. [2025c]).

**Discussion: Rethinking as Perceptual Verification**. A primary takeaway from our work is the qualitative difference in "rethinking" between text-only and vision-language models. In our analysis, we found no strong correlation between improved accuracy and the length of the generated reasoning chain. This suggests that the "slow-thinking" fostered by VL-Rethinker does not improve performance by significantly enhancing logical deliberation, but about improving perceptual accuracy. The model does not learn to think "longer"; it learns to "look twice." The "rethinking" tokens are used to trigger a re-evaluation of the visual input against the initial reasoning step. This insight reframes the challenge of VLM reasoning: the primary bottleneck in the current stage is often not only the reasoning itself, but the synergy of the visual perception and reasoning.

| Model | Math-Related | | | Multi-Discipline | | | Real-World |
|---|---|---|---|---|---|---|---|
| | MathVista testmini | MathVerse testmini | MathVision test | MMMU-Pro overall | MMMU val | EMMA full | MEGA core |
| *Proprietary Model* | | | | | | | |
| OpenAI-o1 | 73.9 | 57.0 | 60.3 | 62.4 | 78.2 | 45.7 | 56.2 |
| OpenAI-GPT-4o | 60.0 | 41.2 | 30.6 | 51.9 | 69.1 | 32.7 | 52.7 |
| Claude-3.5-Sonnet | 67.7 | 47.8 | 33.5 | 51.5 | 68.3 | 35.1 | 52.3 |
| Gemini-2.0-Flash | 73.4 | 54.6 | 41.3 | 51.7 | 70.7 | 33.6 | 54.1 |
| *Open-Source Models* | | | | | | | |
| Llama4-Scout-109B | 70.7 | - | - | 52.2 | 69.4 | 24.6 | 31.8 |
| InternVL-2.5-78B | 72.3 | 51.7 | 34.9 | 48.6 | 61.8 | 27.1 | 44.1 |
| QvQ-72B | 71.4 | 48.6 | 35.9 | 51.5 | [†]66.7 | 32.0 | 8.8 |
| LLava-OV-72B | 67.5 | 39.1 | 30.1 | 31.0 | 56.8 | 23.8 | 29.7 |
| Qwen2.5-VL-32B | 74.7 | 48.5 | 38.4 | 49.5 | [†]59.4 | 31.1 | 13.3 |
| Qwen2.5-VL-72B | 74.8 | 57.2 | 38.1 | 51.6 | [†]67.0 | 34.1 | 49.0 |
| VL-Rethinker-32B | 78.8 | 56.9 | 40.5 | 50.6 | 65.6 | 37.9 | 19.9 |
| VL-Rethinker-72B | **80.4** | **63.5** | **44.9** | **55.9** | **68.8** | **38.5** | **51.3** |
| Δ (Ours - Open SoTA) | +5.6 | +6.3 | +6.5 | +3.7 | -0.6 | +4.4 | +2.3 |

Table 1: Comparison between our 72B model and other state-of-the-art models. The notation of [†] indicates reproduced results using our evaluation protocols.

# 4 Experiments

In this section, we first outline the training and evaluation settings, and then examine the key factors for effectively fostering deliberate reasoning in VLMs.

**Training Data and Benchmarks.** Our training data was compiled by integrating publicly available datasets [Du et al., 2025, Yang et al., 2025, Meng et al., 2025] with novel data collected from the web. This initial "seed" query set underwent a rigorous cleaning and augmentation pipeline, yielding a high-quality dataset of approximately 38,870 queries. Analysis of training dynamics (Fig. 2) revealed that RL training on the seed queries quickly reached saturation, so we strategically curated different query subsets for training models of varying scales based on query difficulty. This procedure resulted in specialized subsets: approximately 16,000 queries for 7B model training and 20,000 queries for 32B and 72B model training, representing a spectrum of performance levels for each corresponding model. A detailed description of our data preparation methodology is provided in the appendix.

For evaluation, we employ a diverse set of challenging multimodal benchmarks:

- Math-related reasoning: MathVista [Lu et al., 2023], MathVerse [Zhang et al., 2024], and MathVision [Wang et al., 2024a].
- Multi-discipline understanding and reasoning: MMMU [Yue et al., 2024a], MMMU-Pro [Yue et al., 2024b], and EMMA [Hao et al., 2025].
- Large-scale long-tailed real-world tasks: MegaBench [Chen et al., 2024a].

This benchmark suite covers a wide range of complex multimodal reasoning challenges. We report the Pass@1 accuracy using greedy decoding.

**Baselines and Implementation.** We compare against several categories of models:

- Proprietary models: GPT-4o [Hurst et al., 2024], o1 [Jaech et al., 2024], Claude 3.5 Sonnet [Anthropic, 2024], Gemini-2.0-Flash [Team et al., 2023].
- State-of-the-art open-source models: Qwen2.5-VL-72B [Bai et al., 2025], QvQ-72B [Wang et al., 2024b], InternVL-2.5-78B [Chen et al., 2024b], Llava-Onevision [Li et al., 2024], Llama-4-Scout and Kimi-VL [Team et al., 2025].
- Representative open-source reasoning-focused models: OpenVLThinker [Deng et al., 2025], R1-OneVision [Yang et al., 2025], R1-VL [Zhang et al., 2025] and MM-Eureka [Meng et al., 2025]. These models are mainly trained on multimodal reasoning dataset.

Our VL-Rethinker-72B was trained using OpenRLHF for a maximum of 3 epochs on 8 sets of $8 \times A800(80G)$ for approximately 60 hours. We include training details in the appendix, and will release code, models and our high-quality 39K dataset.

| Model | Math-Related | | | Multi-Discipline | | | Real-World |
|---|---|---|---|---|---|---|---|
| | MathVista testmini | MathVerse testmini | MathVision test | MMMU-Pro overall | MMMU val | EMMA full | MEGA core |
| General Vision-Language Models | | | | | | | |
| InternVL2-8B | 58.3 | - | 17.4 | 29.0 | 51.2 | 19.8 | 26.0 |
| InternVL2.5-8B | 64.4 | 39.5 | 19.7 | 34.3 | 56.0 | - | 30.4 |
| Qwen2-VL-7B | 58.2 | - | 16.3 | 30.5 | 54.1 | 20.2 | 34.8 |
| Qwen2.5-VL-7B | 68.2 | 46.3 | 25.1 | 36.9 | †54.3 | 21.5 | 35.0 |
| Llava-OV-7B | 63.2 | 26.2 | - | 24.1 | 48.8 | 18.3 | 22.9 |
| Kimi-VL-16B | 68.7 | 44.9 | 21.4 | - | †55.7 | - | - |
| Vision-Language Reasoning Models | | | | | | | |
| MM-Eureka-8B (InternVL) | 67.1 | 40.4 | 22.2 | 27.8 | 49.2 | - | - |
| R1-VL-7B | 63.5 | 40.0 | 24.7 | 7.8 | 44.5 | 8.3 | 29.9 |
| R1-Onevision-7B | 64.1 | 46.4 | 29.9 | 21.6 | - | 20.8 | 27.1 |
| OpenVLThinker-7B | 70.2 | 47.9 | 25.3 | 37.3 | 52.5 | 26.6 | 12.0 |
| VL-Rethinker-7B | **74.9** | **54.2** | **32.3** | **41.7** | **56.7** | **29.7** | **37.2** |
| Δ (Ours - Prev SoTA) | +4.7 | +6.3 | +2.4 | +4.4 | +0.7 | +3.1 | +2.2 |
| Ablations (Incrementally Ablated from VL-Rethinker-7B) | | | | | | | |
| VL-Reasoner-7B | 72.4 | 53.2 | 29.8 | 40.9 | - | 29.5 | - |
| w/o SSR (=DAPO) | 72.0 | 50.0 | 28.5 | 40.0 | - | 26.9 | - |
| w/o Filter (=GRPO) | 71.2 | 50.8 | 27.4 | 39.2 | - | 26.4 | - |

Table 2: Comparison between our 7B model and other general and reasoning vision-language models. † means that the results are reproduced by us.

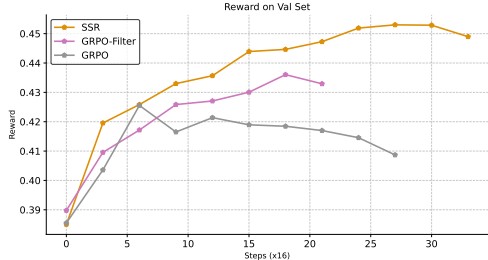

Figure 5: Comparisons of training dynamics of GRPO, GRPO-Filter and GRPO-SSR. GRPO baseline exhibits significant overfit, and GRPO-Filter are more stabilized. GRPO-SSR achieves the best convergence.

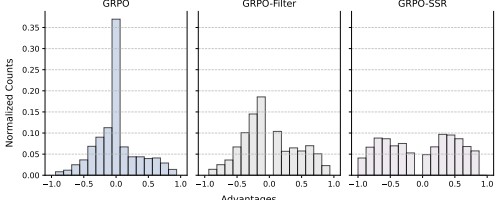

Figure 6: Comparisons of training batch advantage distribution. Standard GRPO and GRPO-Filter has biased advantage distribution, with mass centered around zero. In contrast, GRPO-SSR re-distribute the probability mass over training examples evenly across different advantage values.

## 4.1 Main Results

Our approach demonstrates significant performance gains, as evidenced by the quantitative results. For the 72B models (Table 1), VL-Rethinker-72B achieved significant improvements over the base model, Qwen2.5-VL-72B. Notably, VL-Rethinker-72B achieved state-of-the-art results on math-related benchmarks among all models, including OpenAI-o1. For the 7B models (Table 2), VL-Rethinker-7B outperforms competitor 7B models that also employ RL, e.g., OpenVLThinker, R1-OneVision, by a large margin. These results underscore the effectiveness of our proposed approach in enhancing performance across various challenging benchmarks.

## 4.2 Ablation Studies

**Ablation on Data.** We include an ablation of data compositions in the appendix.

**Ablation on Selective Sample Replay (SSR).** To address vanishing advantages, we introduce Selective Sample Replay (SSR) based on GRPO. GRPO-SSR filters out queries causing zero advantages and perform selective sampling with a probability proportional to the absolute advantage. To investigate the impact of filtering and selective replay, we establish two corresponding baselines for comparison against VL-Reasoner (a baseline without "Forced Rethinking"): (a) the baseline without selective replay but retains the filtering, denoted as GRPO-Filter (also used in DAPO [Yu et al., 2025]); (b) the baseline with neither sample replay nor filtering, identical to standard GRPO.

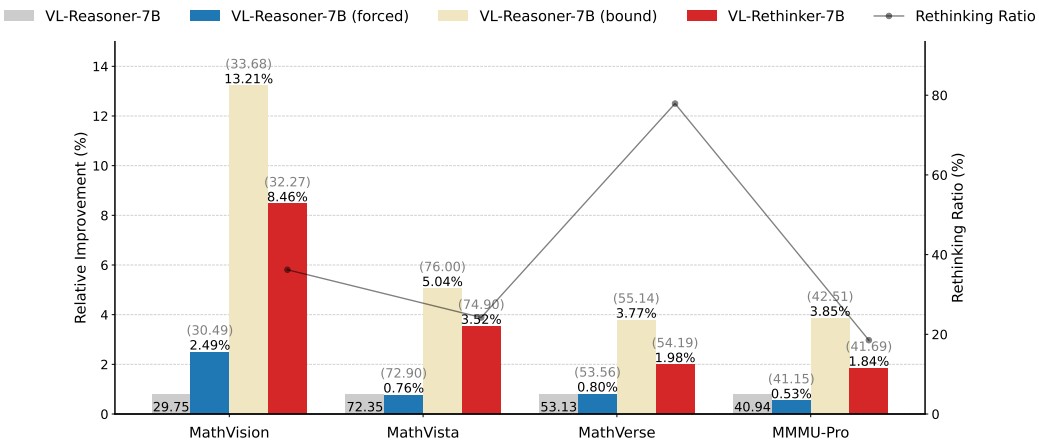

Figure 7: **Relative Improvement with Different Re-thinking Strategies.** We compare: (a) VL-Reasoner (forced), which is forced to rethink at test time; (b) VL-Reasoner (bound), represents the upper bound of test-time forced re-thinking; and (c) VL-Rethinker is trained for self-reflection. The results indicate that forcing VL-Reasoner to rethink at test time yields positive performance gains. Training for self-reflection significantly enhances performance, achieving closer results to the upper bound of forced re-thinking. The overlaid line plot shows the rethinking ratio (right y-axis) of VL-Rethinker across different benchmarks, showing VL-Rethinker adaptively performs re-thinking, unlike the fixed forced re-thinking strategy.

The results presented in Table. 2 highlight the effectiveness of our proposed components. The models trained with the full GRPO-SSR algorithm consistently achieves superior performance compared to the ablated versions, strongly supporting the benefits of both filtering and selective replay.

Further insights into the behavior of these algorithms are revealed by analyzing the training dynamics, as shown in Fig. 5. the GRPO baseline exhibits the most pronounced overfitting, eventually leading to performance degradation. This can be attributed to the vanishing advantages problem, where the number of training examples with near-zero advantages increases as training progresses. These examples provide minimal learning signal, effectively reducing the batch size and destabilizing the training process. In contrast, GRPO-SSR demonstrates a more stable training process and achieves better convergence compared to GRPO-Filter, suggesting the beneficial role of SSR.

The underlying reason for these differences is illuminated by the advantage distributions during training (Fig. 6). Standard GRPO displays a highly skewed distribution, with a pronounced peak at zero advantage, confirming that a large fraction of samples provides ineffective gradients. GRPO-Filter alleviates the extreme peak at zero, yet it still retains a strong central bias, indicating that many examples with very small advantages persist.

Conversely, GRPO-SSR significantly alters the advantage distribution by redistributing the probability mass away from zero and placing greater emphasis on examples with large absolute advantages. These examples, such as a correct response to a challenging query or an incorrect response to a simple one, are intuitively more informative as they likely lie closer to the decision boundary. By selectively replaying these high-advantage examples, GRPO-SSR ensures a more balanced and effective learning process, ultimately leading to improved convergence as evidenced by the reward curves.

**Analysis on Forced Rethinking.** To evaluate the effectiveness of our Forced Rethinking training technique in fostering deliberate reasoning, we compared its impact against baseline models and theoretical limits, as illustrated in Fig. 7. Our primary objective was to examine whether training with Forced Rethinking encourages VL-Rethinker to develop internal metacognitive awareness, enabling it to strategically decide when rethinking is beneficial, rather than applying it rigidly.

Fig. 7 compares the performance of VL-Rethinker against several configurations. The baseline is "w/o Forced Rethinking", which we dub *VL-Reasoner*. We first assessed the inherent potential of rethinking via *VL-Reasoner (forced)*, where the baseline model is compelled to perform a rethinking step at test time for every instance. The results (blue bars) show positive relative improvements across all benchmarks. This indicates that the baseline model already possesses latent rethinking capabilities

that can lead to correct answers. However, this approach is suboptimal, as the baseline struggles to effectively leverage this ability, sometimes even corrupting initially correct answers through flawed rethinking. We also compute an upper bound, *VL-Reasoner (bound)* (yellow bars), which represents the maximum achievable improvement if test-time rethinking is only applied to the wrong outputs.

Crucially, `VL-Rethinker` (red bars), trained using our Forced Rethinking technique, consistently outperforms the *VL-Reasoner (forced)* baseline. For example, on MathVision, *VL-Rethinker* achieves an 8.46% relative improvement, significantly higher than the 2.49% gained by passively forcing the baseline to re-think. This demonstrates that integrating rethinking into the training phase markedly enhances the model's capacity for effective self-reflection.

Importantly, the analysis highlights the adaptive nature of the learned rethinking behavior. The overlaid line plot (right y-axis) shows the "Rethinking Ratio" for `VL-Rethinker` – the fraction of test instances where it spontaneously engaged in the rethinking process. This ratio varies substantially across benchmarks, in stark contrast to the rigid, 100% application in the *VL-Reasoner (forced)* scenario. It suggests that `VL-Rethinker` has learned to selectively trigger re-thinking based on the query's perceived difficulty or its initial confidence, embodying the targeted metacognitive awareness rather than relying on a fixed, potentially inefficient strategy.

## 5  Related Work

### 5.1  Multimodal Instruction Tuning

Instruction tuning has become a central technique for aligning large language models (LLMs) with human intent, enabling them to better follow open-ended natural language instructions. In the multimodal setting, however, aligning both language and vision modalities presents unique challenges. Building upon the success of unimodal instruction tuning methods such as FLAN [Wei et al., 2022], Self-Instruct [Wang et al., 2023], and Direct Preference Optimization (DPO) [Rafailov et al., 2023], researchers have extended these strategies to vision-language models (VLMs). These models must reason over visual semantics, resolve cross-modal references, and produce grounded, coherent responses—all within the framework of natural language instructions.

Initial efforts such as InstructBLIP [Dai et al., 2023], LLaVA [Liu et al., 2023], and MiniGPT-4 [Zhu et al., 2024] demonstrated the feasibility of aligning VLMs using instruction-following data. More recent advances, including Llava-OV [Li et al., 2024], Infinity-MM [Gu et al., 2024], MAmmoTH-VL [Guo et al., 2024], and VisualWebInstruct [Jia et al., 2025], show that scaling up instruction tuning datasets and introducing diverse tasks can significantly enhance generalization across a wide range of multimodal benchmarks.

### 5.2  Reasoning with Reinforcement Learning

The release of GPT-o1 [Jaech et al., 2024] and DeepSeek-R1 [Guo et al., 2025] has sparked renewed interest in incentivizing reasoning capabilities in LLMs via reinforcement learning (RL). Recent works like SimpleRL-Zoo [Zeng et al., 2025] and Open-Reasoner-Zero [Hu et al., 2025] explore direct RL fine-tuning from base models without relying on additional supervised instruction-tuning phases. Building on this foundation, approaches such as DeepScaler [Luo et al., 2025] and Light-R1 [Wen et al., 2025] incorporate cold-start datasets specifically designed to promote long-form reasoning and step-by-step thought processes.

In parallel, efforts such as DAPO [Yu et al., 2025] and Dr GRPO [Liu et al., 2025] aim to improve the original Group Relative Policy Optimization (GRPO) algorithm, refining reward structures and advantage estimation to more effectively elicit deep reasoning behaviors from LLMs during training.

### 5.3  Multimodal Reinforcement Learning

There is a growing body of work focused on bringing RL-based reasoning into the multimodal domain [Deng et al., 2025, Yang et al., 2025, Huang et al., 2025, Peng et al., 2025]. Inspired by models like DeepSeek-R1, these approaches typically follow a multi-stage pipeline. A common practice involves first performing supervised fine-tuning (SFT) on vision-language data that has been annotated or augmented with detailed reasoning traces, often derived from strong text-only LLMs after converting visual inputs into textual descriptions.

Following the SFT stage, reinforcement learning is used to further enhance the model's reasoning capabilities. While effective, these pipelines often require complex and resource-intensive processes, including visual captioning, teacher model distillation, and tightly coupled SFT+RL orchestration [Wang et al., 2025c]. In contrast, our work investigates a more direct and lightweight RL-only approach, aiming to incentivize slow-thinking behavior without relying on large-scale supervision or teacher-based distillation.

# 6 Conclusion

In this paper, we investigated how to more effectively incentivize the reasoning capabilities of multimodal models. Our proposed approaches have shown effectiveness in multimodal reasoning benchmarks. However, our models are still lagging behind human expert performance on more general multimodal tasks like EMMA and MEGA-Bench. We conjecture that this is due to a lack of high-quality multimodal training dataset. In the future, we endeavor to further improve the data quality to improve multimodal reasoning capabilities.

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

# Appendix

## A  Limitations and Discussions

In this work, we studied how to effectively cultivate slow-thinking via Reinforcement Learning. We proposed Selective Sample Replay to mitigate the vanishing advantages in GRPO, and employed forced rethinking to foster deliberate reasoning. While we achieve state-of-the-art results on math-related benchmarks, our models still lag behind human expert performance on more general multimodal benchmarks like EMMA and MEGA-Bench. This reveals that our model is still limited in high-quality training queries. While we show that a direct RL approach without costly distillation can outperform existing RL-based VLMs that involve costly distillations, it remains an open question in what conditions SFT can indeed help the subsequent RL phase for VLMs.

## B  Training and Implementations

### B.1  Training Dataset

Our initial seed query set was constructed by aggregating publicly available multimodal datasets [Yang et al., 2025, Meng et al., 2025, Kembhavi et al., 2016, Saikh et al., 2022, Du et al., 2025] with novel queries gathered from the web. This aggregated dataset exhibits a broad topical diversity, as visually represented in Fig. 8. Given our reliance on rule-based reward mechanisms for subsequent Reinforcement Learning (RL) training, a crucial first step involved filtering the seed queries. We retained only those queries with reference answers that were programmatically verifiable by our defined rules. From this verifiable subset, an augmented query set was systematically generated through the rephrasing of questions and permutation of multi-choice options. This augmentation strategy was designed to facilitate knowledge re-occurrence and reinforce learning across variations of the same core information. This rigorous data preparation pipeline culminated in a final training set comprising 38,870 queries.

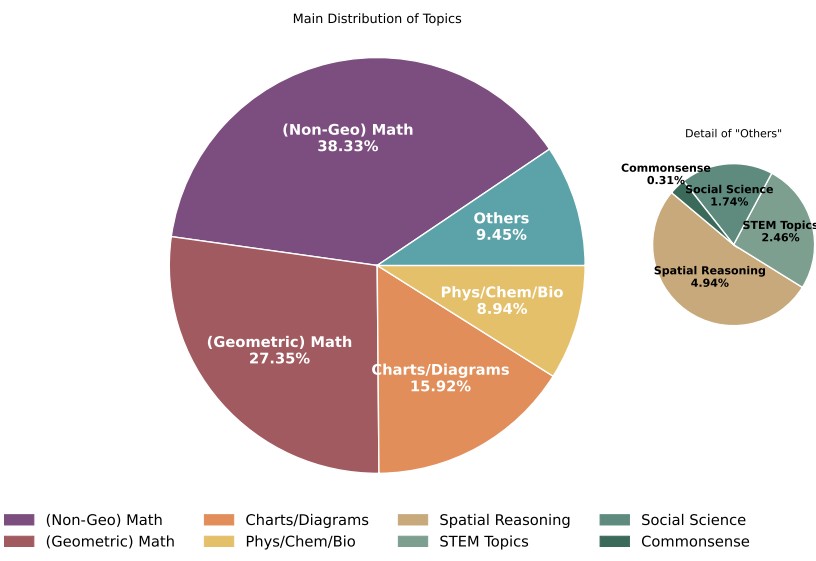

Figure 8: **Our training data contains a diverse collection of topics, including eight major categories.**

Utilizing this comprehensive query set, we proceeded to train models at different scales. To ensure efficient training and leverage each model's inherent strengths, we selected subsets of queries tailored to their initial capabilities. Specifically, for each model scale, we curated a training subset consisting of queries where the initial checkpoint of that model demonstrated a non-zero PassRate@8. This selection criterion ensured that the models were trained on queries falling within their potential

---

**Algorithm 1** Selective Sample Replay (SSR)

---

1: **Input:** Buffer $\mathcal{B}_{\text{replay}}$, raw training batch $\mathcal{D}_{\text{raw}} = \{(x_i, y_i, \hat{A}_i)\}$, intensity $\alpha \geq 0$.
2: **Output:** Training batch $\mathcal{D}_{\text{train}}$, updated buffer $\mathcal{B}_{\text{replay}}$
3: Let $N_{\text{batch}} = |\mathcal{D}_{\text{raw}}|$
4: Initialize list for effective current samples $\mathcal{D}_{\text{effective}} \leftarrow \emptyset$
5: **for** each sample $(x_i, y_i, \hat{A}_i)$ in $\mathcal{D}_{\text{raw}}$ **do**
6:     Add $(x_i, y_i, \hat{A}_i)$ to $\mathcal{D}_{\text{effective}}$ when $|\hat{A}_i| > 0$
7: **end for**
8: Update buffer: $\mathcal{B}_{\text{replay}} \leftarrow \mathcal{B}_{\text{replay}} \cup \mathcal{D}_{\text{effective}}$
9: Let $n_{\text{effective}} = |\mathcal{D}_{\text{effective}}|$
10: Calculate number of samples needed from buffer: $n_{\text{from\_buffer}} = \max(0, N_{\text{batch}} - n_{\text{effective}})$
11: Initialize list for samples from buffer $\mathcal{D}_{\text{from\_buffer}} \leftarrow \emptyset$
12: **if** $n_{\text{from\_buffer}} > 0$ **then**
13:     Calculate sampling probabilities $P(\text{select } j)$ for all $j \in \mathcal{B}_{\text{replay}}$ according to Eq. 1
14:     Form $\mathcal{D}_{\text{from\_buffer}}$ by drawing $n_{\text{from\_buffer}}$ samples from $\mathcal{B}_{\text{replay}}$
15: **end if**
16: $\mathcal{D}_{\text{train}} \leftarrow \mathcal{D}_{\text{effective}} \cup \mathcal{D}_{\text{from\_buffer}}$

---

competence range, allowing the RL process to refine and enhance existing, albeit nascent, abilities rather than attempting to instill knowledge from scratch.

## B.2 Algorithms

We provide a diagram for Selective Sample Replay in Alg. 1

## B.3 Implementations and Training Details

Our `VL-Rethinker-72B` was trained using OpenRLHF for a maximum of 3 epochs on 8 sets of $8 \times \text{A800(80G)}$ for approximately 60 hours. The final checkpoint was selected based on the mean reward achieved on a held-out validation set. We employed a near on-policy RL paradigm, where the behavior policy was synchronized with the improvement policy after every 1024 queries, which we define as an episode. The replay buffer for SSR persisted for the duration of each episode before being cleared. For each query, we sampled 8 responses. The training batch size was set to 512 query-response pairs. We accept at most two correct rethinking trajectories for each query. We set the priority hyperparameter in SSR to $\alpha = 1.0$ in the experiments. We released code, models and our high-quality 39K dataset to support further research.

## B.4 Prompts Used for Training

---

**Default Instruction Prompt**

```
{question}
Please reason step by step, and put your final answer within \\boxed{}.
```

During the first stage RL training with SSR, we use the default instruction prompt as above.

---

**Rethinking Instruction Prompt**

```
{question}
Guidelines:

Please think step by step, and **regularly perform self-questioning, self
-verification, self-correction to check your ongoing reasoning**, using
connectives such as "Wait a moment", "Wait, does it seem right?", etc. Remember
to put your final answer within \\boxed{}.
```

During the Forced Rethinking training stage, we use the above prompt to encourage self-reflection, and use three types of rethinking textual triggers.

```
self_questioning = "\n\nWait, does it seem right?"
self_correction = "\n\nWait, there might be a mistake"
self_verification = "\n\nWait, let's double check"
```

## C   Additional Experiments and Analysis

### C.1   Experiments

We conducted an ablation on the data compositions. Our training queries are comprised of three major genres: math-related vision-language queries, science-related queries and text-only ones. We conducted ablation studies on these components. As shown in Table. 3, removing text-only queries does not cause significant differences. As we further remove queries from the broader scientific domains, we observe a more pronounced drop in performance. This significant reduction underscores the importance of scientific data in improving the model's general reasoning ability.

| **Model** | RL-Algo | Data | MathVision | MathVista | MathVerse | MMMU-Pro | EMMA |
|-----------|---------|------|-----------|-----------|-----------|----------|------|
| VL-Rethinker-7B | SSR | 16K | 32.3 | 74.9 | 54.2 | 41.7 | 29.7 |
| w/o 'Forced-Rethinking' | SSR | 16K | 29.8 | 72.4 | 53.2 | 40.9 | 29.5 |
| - no SSR | Filter | 16K | 28.5 | 72.0 | 50.0 | 40.0 | 26.9 |
| - no SSR& Filter | GRPO | 16K | 26.0 | 70.9 | 51.4 | 38.8 | 26.2 |
| - no Text | SSR | 13K | 29.1 | 73.5 | 53.5 | 41.1 | 28.7 |
| - no Science&Text | SSR | 11K | 28.0 | 71.6 | 50.3 | 39.7 | 28.0 |

Table 3: Ablation Results to show the impact of SSR and Data Mix.

We performed a sensitivity analysis to the prioritization intensity hyperparameter in SSR. We plot the final averaged eval reward of models trained with different hyperparameters. As shown in Fig. 9, the performance does not vary significantly regarding different hyperparameters, and the best value is around 1.0. We observe that models trained with different hyperparameters will always reach similar peaks sooner or later.

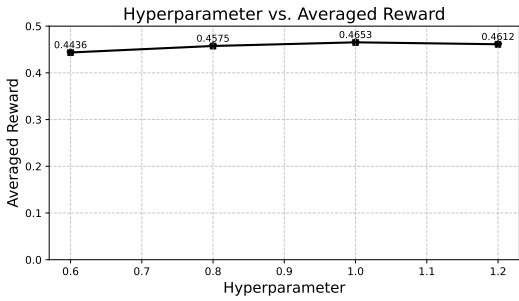

Figure 9: Sensitivity Analysis of the Hyperparameter in SSR

### C.2   Cases

Below we show a few cases where VL-Rethinker shows the ability of rethinking. This ability can help achieve correct answers more often than not, but sometimes it still fails with challenging questions.

Listing 1: successful case1

```
Q: Hint: Please answer the question requiring an integer answer and
    ↪ provide the final value, e.g., 1, 2, 3.
What is the highest amount this class measures?
GT: \boxed{600}
A: To determine the highest amount this beaker measures, let's analyze
    ↪ the image step by step:

1. The beaker is labeled \"600ml\" at the bottom, which indicates its
    ↪ total capacity.

2. The markings on the side of the beaker show volume levels: 100, 200,
    ↪ 300, and 400 ml.

3. The highest marking visible on the beaker is 400 ml.

Wait a moment, does the highest marking represent the total capacity?
    ↪ No, it doesn't. The beaker is labeled \"600ml\" at the bottom,
    ↪ which means it can hold up to 600 ml of liquid.

Wait, does it seem right? Yes, it does. The beaker is designed to hold
    ↪ 600 ml, even though the highest marking is 400 ml. The remaining
    ↪ capacity above 400 ml is not marked but is still part of the
    ↪ beaker's total capacity.

Therefore, the highest amount this beaker measures is 600 ml.
\\boxed{600}
```

Listing 2: successful case2

```
Q: In the triangle with hypotenuse length 20, find
    ↪ y.\nChoices:\nA:4\nB:6\nC:8\nD:10
GT: \boxed{D}
A: To find the length of \\( y \\) in the given right triangle with a
    ↪ hypotenuse of 20 and an angle of 60\u00b0, we can use
    ↪ trigonometric relationships.
1. **Identify the trigonometric relationship:**
- In a right triangle, the sine of an angle is the ratio of the length
    ↪ of the opposite side to the hypotenuse.
- Here, \\( \\sin(60\u00b0) = \\frac{y}{20} \\).

2. **Recall the value of \\( \\sin(60\u00b0) \\):**
- \\( \\sin(60\u00b0) = \\frac{\\sqrt{3}}{2} \\).

3. **Set up the equation:**
\\[\n   \\sin(60\u00b0) = \\frac{y}{20} \\implies \\frac{\\sqrt{3}}{2} =
    ↪ \\frac{y}{20}\n   \\]\n\n4. **Solve for \\( y \\):**\n   \\[\n   y
    ↪ = 20 \\cdot \\frac{\\sqrt{3}}{2} = 10\\sqrt{3}\n   \\]

**Wait a moment**, let's double check the problem statement and the
    ↪ choices.** The problem asks for \\( y \\) and the choices are 4,
    ↪ 6, 8, 10. It seems there might be a mistake in the problem setup
    ↪ or the choices provided. However, based on the trigonometric
    ↪ calculation, the correct value of \\( y \\) is \\( 10\\sqrt{3}
    ↪ \\), which is approximately 17.32, not matching any of the given
    ↪ choices.

Given the choices, the closest integer value to \\( 10\\sqrt{3} \\) is
    ↪ 10, but it is not exactly 10.

**Wait, does it seem right?** The problem might have a different context
    ↪ or the choices might be incorrect.

Since the problem asks for the closest value and 10 is the closest to
    ↪ \\( 10\\sqrt{3} \\), the answer is:\n\n\\boxed{D}
```

```
Q: In the figure there are nine regions inside the circles. The numbers
    ↪ 1 to 9 should be written in the regions so that the sum of the
    ↪ numbers in each circle is exactly 11. Which number has to go in
    ↪ the region with the question mark?<image1>

GT: \boxed{6}
A: To solve this problem, we need to place the numbers 1 to 9 in the
    ↪ nine regions such that the sum of the numbers in each circle is
    ↪ exactly 11. Let's denote the regions as follows:

- \\( A \\) is the region in the top left circle.

- \\( B \\) is the region in the top middle circle.

- \\( C \\) is the region in the top right circle.

- \\( D \\) is the region in the bottom left circle.

- \\( E \\) is the region in the bottom middle circle.

- \\( F \\) is the region in the bottom right circle.

- \\( G \\) is the region in the middle left circle.

- \\( H \\) is the region in the middle right circle.

- \\( I \\) is the region in the middle circle (the one with the
    ↪ question mark).

The circles and their regions can be described as follows:
- The top left circle contains \\( A, G, D \\).
- The top middle circle contains \\( G, B, H \\).
- The top right circle contains \\( H, C, F \\).
- The bottom left circle contains \\( D, E, I \\).
- The bottom right circle contains \\( I, F, E \\).

We need to satisfy the following equations:
1. \\( A + G + D = 11 \\)
2. \\( G + B + H = 11 \\)
3. \\( H + C + F = 11 \\)
4. \\( D + E + I = 11 \\)
5. \\( I + F + E = 11 \\)

Let's start by adding all five equations:
\\[ (A + G + D) + (G + B + H) + (H + C + F) + (D + E + I) + (I + F + E)
    ↪ = 55 \\]
This simplifies to:
\\[ A + B + C + 2D + 2E + 2F + 2G + 2H + 2I = 55 \\]
Since \\( A + B + C + D + E + F + G + H + I = 45 \\) (the sum of the
    ↪ numbers 1 to 9), we can substitute \\( 45 + D + E + F + G + H + I
    ↪ = 55 \\), which simplifies to:
\\[ D + E + F + G + H + I = 10 \\]

Now, let's try to find a solution by trial and error, keeping in mind
    ↪ that each circle must sum to 11.

Assume \\( I = 1 \\):
```

Then $D + E + 1 = 11$ and $1 + F + E = 11$, so $D + E = 10$ and $F + E = 10$. This means $D = F$, which is not possible since all numbers must be distinct.

Assume $I = 2$:
Then $D + E + 2 = 11$ and $2 + F + E = 11$, so $D + E = 9$ and $F + E = 9$. This means $D = F$, which is not possible.

Assume $I = 3$:

Then $D + E + 3 = 11$ and $3 + F + E = 11$, so $D + E = 8$ and $F + E = 8$. This means $D = F$, which is not possible.

Assume $I = 4$:
Then $D + E + 4 = 11$ and $4 + F + E = 11$, so $D + E = 7$ and $F + E = 7$. This means $D = F$, which is not possible.

Assume $I = 5$:
Then $D + E + 5 = 11$ and $5 + F + E = 11$, so $D + E = 6$ and $F + E = 6$. This means $D = F$, which is not possible.

Assume $I = 6$:
Then $D + E + 6 = 11$ and $6 + F + E = 11$, so $D + E = 5$ and $F + E = 5$. This means $D = F$, which is not possible.

Assume $I = 7$:
Then $D + E + 7 = 11$ and $7 + F + E = 11$, so $D + E = 4$ and $F + E = 4$. This means $D = F$, which is not possible.

Assume $I = 8$:
Then $D + E + 8 = 11$ and $8 + F + E = 11$, so $D + E = 3$ and $F + E = 3$. This means $D = F$, which is not possible.

Assume $I = 9$:
Then $D + E + 9 = 11$ and $9 + F + E = 11$, so $D + E = 2$ and $F + E = 2$. This means $D = F$, which is not possible.

After trying all possibilities, we find that the only number that works is $I = 1$.

**Wait, does it seem right? Let's verify:**
If $I = 1$, then $D + E = 10$ and $F + E = 10$. Let's try $D = 6$ and $E = 4$, and $F = 6$ and $E = 4$. This is not possible. Let's try $D = 5$ and $E = 5$, and $F = 5$ and $E = 5$. This is not possible. Let's try $D = 3$ and $E = 7$, and $F = 3$ and $E = 7$. This is not possible. Let's try $D = 2$ and $E = 8$, and $F = 2$ and $E = 8$. This is not possible. Let's try $D = 1$ and $E = 9$, and $F = 1$ and $E = 9$. This is not possible.

After all these trials, we find that the only number that works is $I = 1$.

Therefore, the number that has to go in the region with the question mark is $\boxed{1}$.

