# OpenReview forum: "VL-Rethinker: Incentivizing Self-Reflection of Vision-Language Models with Reinforcement Learning"
_NeurIPS.cc/2025/Conference — NeurIPS 2025 spotlight_

### Official Review · Reviewer_EyWX · 2025-07-02

**Clarity:** 3
**Significance:** 3
**Originality:** 3
**Rating:** 5
**Confidence:** 4

**Summary:**

1. The paper introduces VL-Rethinker, a novel framework to train vision-language models (VLMs) for adaptive, reflective, and step-by-step reasoning using direct reinforcement learning—without any supervised fine-tuning, chain-of-thought supervision, or teacher model distillation. The approach is built on Qwen2.5-VL-Instruct (7B, 32B, 72B scales).
2. A key novelty is the Forced Rethinking mechanism, where natural language triggers like “Wait, is this right?” are inserted into the model’s own generation to initiate a reflective second-stage response, which is reinforced only if the final answer improves.
3. To enable stable and effective RL training in this long-form, sparse-reward setting, the authors introduce Group Relative Policy Optimization (GRPO) with Selective Sample Replay (SSR), which reuses high-advantage trajectories to boost learning efficiency and policy robustness.
4. VL-Rethinker learns to rethink selectively—invoking reflection when needed and skipping it otherwise—and often revises incorrect reasoning steps into correct ones, demonstrating emergent metacognitive behavior.
5. The model is trained on a curated set of ~39K rewardable multimodal QA examples sourced from MathVista, MathVerse, EMMA, MEGA, and other datasets, using programmatic reward functions based solely on final answer correctness.
6. Evaluation is conducted via greedy decoding (no sampling or self-consistency) across 7 benchmarks, including: Math-heavy: MathVista, MathVerse, MathVision, Scientific/diagram: MMMU, EMMA, General open-ended: MEGA. Accuracy is measured via pass@1 based on boxed final answers.
7. VL-Rethinker achieves state-of-the-art open-source performance, outperforming or matching teacher-distilled baselines like OpenVLThinker and R1-OneVision, with the largest gains on math and diagram reasoning tasks. The model shows limited but consistent improvements on scientific (EMMA, MMMU) and real-world (MEGA) tasks, indicating potential for generalization, though the largest gains are in structured tasks with deterministic rewards.
8. The paper releases pretrained models (7B, 32B, 72B) and training data publicly, offering a reproducible and scalable framework for studying self-reflection and direct RL in both vision - language and potentially text-only reasoning models.

**Questions:**

1. Can the authors provide a measure of correctness of reflective steps  - quantify when the model chooses to reflect, how often it helps, and whether reflective steps are more accurate than initial ones? Statistics on reflection frequency vs. task difficulty, and success rates of revised answers compared to original answers, would support claims of emergent metacognition.
2. Failure case analysis or reward shaping details could reveal whether the system is robust or prone to unnecessary revision loops. Does rethinking ever lead to degraded answers or hallucinations? How does the model learn to suppress unhelpful reflection?
3.Even a small-scale experiment on a text-only task would help test the modality-agnostic claim and broaden the paper’s applicability.
4. A breakdown of errors corrected through rethinking—separated into vision-specific (e.g., perceptual misinterpretation) and logic-specific (e.g., math or inference errors)—would clarify whether reflection leverages the visual modality meaningfully. It would help to clarify whether the rethinking mechanism helps the model improve visual perception (e.g., object recognition, spatial relationships), or if it primarily corrects symbolic/logical reasoning based on already-processed visual features?

**Ethical Concerns:**

["NO or VERY MINOR ethics concerns only"]

**Final Justification:**

The rebuttal responses clarify some of the important questions including the qualitative analysis of rethinking and rethinking success across multiple difficulty levels and impact of SSR in modality agnostic/ text settings.
The two questions remain open, 1) error analysis of rethinking trigger to correct different categories of error, 2)downside of SSR approach. But, overall the quality of the paper is good and rebuttal response further helps to update the rating.

**Limitations:**

Limitations are fairly discussed in supplementary material.

**Paper Formatting Concerns:**

No major formatting concerns. It would be useful to expand and move limitations section in the main paper in future revisions.

**Quality:**

3

**Strengths And Weaknesses:**

Strengths -
1. The paper proposes a novel direct reinforcement learning (RL) framework to induce reflective, step-by-step reasoning in vision-language models (VLMs) without relying on supervised CoT data, rationales, or teacher model distillation—marking a significant advance in self-improving LLM training.
2. It introduces Forced Rethinking, a simple but effective mechanism using natural-language triggers to teach the model to re-evaluate its own answers, enabling reflective behavior to emerge purely through reward-driven optimization.
3. The method scales robustly across model sizes (7B, 32B, 72B) and works across domains (math, science, diagrams, general QA), demonstrating scalability and domain generality without additional supervision.
4. VL-Rethinker offers a generalizable training recipe that is modality-agnostic in principle, paving the way for reflective reasoning in both vision-language and future text-only LLMs using direct RL approaches.

Areas of Improvement -
1. While Selective Sample Replay (SSR) improves efficiency, the paper does not sufficiently discusses the potential downsides of this approach. It may introduce bias toward early high-reward trajectories, which may reduce exploration and limit the discovery of diverse or novel reasoning strategies.
2. The evaluation primarily focuses on final answer accuracy (pass@1) and does not include direct metrics for rationale quality, correctness of reasoning steps, or effectiveness of reflection—missing an opportunity to quantify reflective behavior.
3. Although the approach could generalize to text-only models, this is not demonstrated—limiting the scope of applicability despite the technique’s modality-agnostic potential.
4. Additionally, VL-Rethinker primarily seems to corrects symbolic or logical errors, which are common in text-only models, but does not clearly demonstrate improvements in vision-specific reasoning such as object grounding, spatial understanding, or visual attribute correction. This raises a question about whether the rethinking mechanism truly leverages visual inputs or merely enhances textual reasoning with such approach. Further error analysis on VL-Rethinker can demonstrate the major error types and patterns, providing direction for further research on visual reasoning.
5. The method relies solely on binary end-of-sequence rewards (i.e., answer correctness), which leads to potential credit misattribution—the RL algorithm may reinforce flawed intermediate reasoning steps that coincidentally lead to the correct final answer. Further analysis on such behavior would be helpful to understand the limitations of proposed approach.

---

> ### Author Rebuttal · Authors · 2025-07-29
>
> We sincerely thank the reviewer for their thorough evaluation and insightful feedback. The comments are highly constructive and have helped us improve the quality of our work.
>
> Below, we conducted additional experiments and analysis to address the questions and provide revision plans for the areas of improvement.
>
> ### **1. On Quantifying Reflective Behavior and Metacognition (Weakness 2, Question 1)**
>
> This is an excellent point. We were aware of this qualitative analysis of reflections but encountered challenges in properly defining reflections and recognizing reflections in the responses: in many occasions, model reflections cannot be detected by pre-defined reflective phrases.
>
> Below we attempted a qualitative analysis by focusing on the reflections triggered by our "forced rethinking". We conducted our analysis using VL-Rethinker-32B inferred responses on the validation data, selecting a subset of queries that keep the correctness of responses balanced. We defined problem difficulty based on the mean accuracy of responses within the same query group. We then measured the frequency of rethinking (i.e., generating the "Wait" trigger), the success rate (initial answer was wrong, final answer is correct), and the failure rate (initial answer was correct, final answer is wrong).
>
> Our findings are summarized in the table below:
>
> | Problem Difficulty (Baseline Accuracy) | Rethinking Frequency | Success Rate of Rethinking | Degradation Rate of Rethinking |
> | :--- | :--- | :--- | :--- |
> | **Easy** ($>75\\%$) | 44.8% | 27.3% | 4.8% |
> | **Medium** ($37.5\\%-75\\%$) | 68.2% | 14.7% | 3.1% |
> | **Hard** ($<37.5\\%$) | 78.5% | 6.9% | 0.8% |
>
> We draw the following key observations:
> * **Adaptive Reflection:** The model’s rethinking frequency is positively correlated with problem difficulty. It adaptively triggers reflection more often on harder problems where its initial answer is more likely to be incorrect.
> * **Effective Rethinking:** While the success rate of rethinking is higher for easier problems and lower for harder problems, rethinking is overall beneficial. The model consistently improves its answers across all difficulty levels via rethinking, with very low degradation rates. The degradation rate is low because the model tends to use self-verification for correct answers, and the outcome reward distinguishes the degraded rethinking response from a beneficial rethinking response.
>
> Here we acknowledge the limitation that problem difficulty is bootstrapped from our model's own performance. We will refine this analysis using more objective judgement in our revision. Nonetheless, these statistics provide quantitative support for our claim of **emergent metacognitive behavior**. The model shows evidence of *when* to rethink and this reflection leads to consistent, positive performance gains.
>
>
> ### **2. On the Modality-Agnostic Claim of SSR (Weakness 3, Question 2)**
>
> We thank the reviewer for asking for clarification on this point. The "vanishing advantages" problem is general in GRPO **when training queries is easy to master**, leading to vast majority of training rollouts being zero advantages. This issue is particularly pronounced for vision-language reasoning domains because there are still a lack of publicly available high-quality challenging reasoning-related queries. In the text-only math reasoning domain, the vanishing advantages problem is mitigated by a few high-quality challenging query dataset, for example, DeepMath. The table below shows the performance improvement on GSM8K and Math500 when training with different collection of training queries (easier-to-master `Math8K` vs. the higher-quality challenging `DeepMath`). The values shown are absolute pass@1 improvements.
>
> | Data | GSM8K | Math500 |
> | :--- | :--- | :--- |
> | Math8K | +1.7 (84.4->89.3 vs 87.6) | +3.2 (60.4->71.4 vs 68.2) |
> | DeepMath | +0.3 (84.4->89.5 vs 89.2) | +1.7 (60.4->78.2 vs 76.5) |
>
> **Key Observations:**
> The benefit of SSR is more pronounced on the `Math8K` dataset. This is because the problems in `Math8K` are easier to master, making the baseline GRPO more susceptible to saturation. On the more uniformly challenging `DeepMath` dataset, the baseline is stronger, but SSR still offers a reliable, albeit smaller, improvement.
>
>
>
> ### **3. On Error Analysis of Rethinking (Question 3)**
> This is a crucial analysis to understand whether rethinking improves visual perception or primarily corrects textual-symbolic reasoning. Due to the time-intensive nature of manually inspecting and annotating numerous generation rollouts, we were unable to complete this analysis during the rebuttal phase.
>
> However, we have a clear plan to incorporate this into the final version of our paper. The analysis will involve manually categorizing corrected errors into groups such as:
>  1.  **Logic-Specific Corrections:** The initial visual perception was correct, but the model corrected a flaw in its mathematical or logical reasoning.
>
>  2.  **Vision-Specific Corrections:** The model initially misinterpreted or failed to locate a key visual element, and the reflection process corrected this perceptual error.
>
>  3.  **Combined Corrections:** Both the visual interpretation and the subsequent logic were flawed and were corrected.
>
>
> ### **4. On the Areas of Improvements**
>
> - **Outcome Reward and Rationale Quality.** We thank the reviewer to pointing out this critical point. Since this work is a preminimary study of slow-thinking in VLMs, the paper follows the prevailing setting of using outcome reward. However we agree this will be a important direction for furture work and we will add this point in the limitation and future work section in the revision.
>
> - **Downsides of SSR.** We agree with the reviewer that discussing the potential downsides of SSR is important. In the meantime, we would like to clarify a detail of our implementation designed to mitigate this risk: SSR prioritizes trajectories based on **absolute advantage**, not just raw rewards. This prevents the policy from merely overfitting to simple, high-reward samples and instead encourages it to focus on trajectories where the policy's performance significantly exceeded its expected value.
>
>     However, we acknowledge that the full impact of SSR on the policy's long-term exploration remains an open area for investigation. We will add a discussion in the paper to elaborate on this potential limitation and frame it as a direction for future research into balancing exploration and exploitation in VLM-based RL.
>
>
> We thank the reviewer once again for their valuable and actionable feedback, which has undoubtedly strengthened our paper.

---

### Official Review · Reviewer_J7nB · 2025-07-03

**Clarity:** 4
**Significance:** 4
**Originality:** 4
**Rating:** 5
**Confidence:** 4

**Summary:**

This work proposes the VL-Rethinker model and associated training method for slow-thinking capabilities in Vision-Language Models. VL-Rethinker uses selective sample replay (SSR) to improve standard GRPO training, which suffers from a vanishing advantages problem due to the advantage normalization. SSR maintains a buffer of past high-value learning experiences. VL-Rethinker also uses forced rethinking, which triggers the model to extend its thinking process during RL training. VL-Rethinker demonstrates strong performance across multimodal reasoning and understanding tasks compared to both open-source and proprietary models. The paper ablates the importance of SSR and forced rethinking and analyzes how they impact the trained model.

**Questions:**

1. How does VL-Rethinker performance vary as a function of the number of thinking tokens per-problem setting?

2. How is the performance impacted by the SSR buffer size?

3. Can forced rethinking work if it is just included in the RL training without the SFT loss? In this setup, the model is trained with GRPO and the SSR technique, but the generated responses used in RL also include the forced rethinking. Is the SFT loss necessary as opposed to just end-to-end RL?

4. How robust is SSR to the choice of alpha? What values of alpha does SSR degrade in performance for? How should the best alpha value be selected?

5. How important is the rethinking fraction and the rethinking prompts to the performance of VL-Rethinker?

**Ethical Concerns:**

["NO or VERY MINOR ethics concerns only"]

**Final Justification:**

Overall, this paper introduces novel techniques in VL-Rethinker for reasoning capabilities in Vision-Language models. VL-Rethinker shows strong empirical performance against baselines. The rebuttal states several additional important analyses will also be added, which will further strengthen the paper.

**Limitations:**

yes

**Quality:**

4

**Strengths And Weaknesses:**

**Strengths**:

1. VL-Rethinker demonstrates strong empirical performance across a variety of competitive benchmarks against many models. VL-Rethinker outperforms the best "slow-thinking" models like OpenAI-o1 and open-source models like QvQ-72B.

1. SSR is a novel technique for fixing the important problem of vanishing advantages in GRPO. Several recent works, such as DAPO [1] and LOOP [2], also observe similar findings for text-based environments. However, SSR is a novel solution that replays high-value data points based on the advantage values. The paper empirically demonstrates this replay-based strategy performs better than a purely filtering-based strategy like DAPO. Furthermore, VL-Rethinker demonstrates this for multimodal tasks.

1. Forced rethinking is also a novel technique for training the model to output reasoning chains. Just like the novelty of SSR, several recent works like S1 have explored similar ideas. However, the forced rethinking technique from this paper is different in that it is applied at training time. As Figure 7 demonstrates, this results in VL-Rethinking being able to adaptively control when it rethinks based on the problem difficulty.

1. The paper has comprehensive analyses ablating the components of VL-Rethinker. First, the paper confirms the vanishing advantages problem, which was previously explored in text-based tasks, also occurs for multimodal tasks. The paper quantifies the impact of the vanishing advantages problem on the effective batch size in Figure 2. Table 2 also confirms the empirical importance of filtering non-meaningful advantages and including the SSR buffer in Table 2. Analysis in Figures 5 and 6 illustrates why these techniques are important to reduce overfitting and prevent a biased advantage distribution. The paper also conducts a similarly detailed investigation for the forced rethinking by comparing to no rethinking or forced rethinking every time. The upper bound of rethinking on only wrong answers also helps ground the results in the possible improvement that can be gained from thinking. This analysis demonstrates how forced rethinking dynamically rethinks based on the problem difficulty.

1. The paper addresses the important problem of reasoning for multimodal tasks, where previously, multimodal reasoning capabilities were similar to non-reasoning models.

1. The paper claims that all the data and training code will be released. The high-quality dataset of 38,870 queries is also valuable for future work to build off of. The paper also provides sufficient implementation details in Appendix B.

**Weaknesses**:

1. The SFT loss for the forced rethinking is not justified. The rest of the training pipeline relies on end-to-end RL, yet the forced rethinking stage requires an additional SFT loss to distill the extended thinking traces. However, this forced rethinking could have also be included in the regular RL training, which is a simpler algorithm and implementation. See my question about the importance of this choice in the "Questions" section.

2. The paper does not include an analysis of the reasoning outputs outside of the rethinking ratio in Figure 7 and qualitative examples in Appendix C.2 Since the paper's primary contribution is improving the reasoning process, it is important to further analyze this. How does the model's performance improve as a function of the number of thinking tokens? How many thinking tokens does the model output for different types of problems, like based on problem difficulty?

3. The paper does not investigate how the size of the SSR buffer impacts performance. For example, a large buffer over a long training run could result in the model learning from stale, very off-policy data, which could negatively impact performance.

4. Figure 9 compares the sensitivity of $\alpha$ in the performance of VL-Rethinker, but does so for a seemingly narrow range of $\alpha$ since the performance of VL-Rethinker is relatively unaffected.

5. The forced rethinking requires several decisions with the selection of the rethinking triggers and rethinking fraction. It's unclear how robust these decisions are to different problem settings.

6. What is the training setting for Figure 2? Is this training the base model on the collected dataset with regular GRPO?

**Minor weaknesses**:

1. What does the bold in Table 1 mean? For MMMU, why is VL-Rethinker bolded, but Llama4-Scout is the best-performing model?

1. I suggest referring to specific sections in the Appendix to make it easier for readers to find the associated sections (L138, L166, L192, L222).

1. There is a missing subscript for $\mathcal{B}_\text{replay}$ on L134.

**References**:

[1] Yu, Qiying, et al. "Dapo: An open-source llm reinforcement learning system at scale." arXiv preprint arXiv:2503.14476 (2025).

[2] Chen, Kevin, et al. "Reinforcement Learning for Long-Horizon Interactive LLM Agents." arXiv preprint arXiv:2502.01600 (2025).

---

> ### Author Rebuttal · Authors · 2025-07-30
>
> We are very grateful to the reviewer for their positive and exceptionally thorough feedback. We are thrilled that they recognized our work as "excellent" and appreciated the novelty of our contributions. Their insightful questions and constructive criticism have been invaluable in helping us identify areas for improvement and strengthen the paper.
>
> Below, we address the weaknesses and questions raised.
>
>
> ### 1. **On the Justification of the SFT Loss**
>
> Thank you for raising this important point. We realize that the role of the auxiliary SFT loss was not sufficiently justified in the manuscript.
>
> The SFT loss is applied concurrently with the RL objective (GRPO) during the training phase. Its primary purpose is to **accelerate the learning of the "forced rethinking" behavior**. While the model could learn to generate the thinking process through the RL objective alone (we tried), we observed that this approach is significantly slower. This is because the trigger tokens for rethinking (e.g., "wait") initially have a very low probability under the policy's distribution. As a result, the discrepancy distribution on the forced tokens is large in the initial training phase and gets clipped. It is until an extended training process that the policy adopts the rethinking tokens. In contrast, the auxiliary SFT loss provides a more direct and stable gradient signal, guiding the policy to adopt the rethinking syntax much more efficiently. We will add a dedicated justification to clarify the motivation for the auxiliary SFT loss in our revision.
>
>
>
> ### 2. **On the Analysis of Reasoning Outputs**
>
> This is an excellent suggestion. We agree that a deeper analysis of the model's reasoning process is crucial to fully demonstrate the benefits of our approach.
>
>   **Revision Plan:** We will add a new analysis section to the Appendix. Due to time constraints during the rebuttal period, we have completed preliminary analyses and will include a comprehensive version in the revision. This new section will include:
>   1.  **Rethinking Efficacy:** A quantitative breakdown of rethinking success and failure rates across different tasks (which we have already completed a preliminary study as per .Reviewer EyWX's suggestions)
>   2.  **Error Analysis:** A comparative error analysis of final answers with and without rethinking to pinpoint the types of mistakes our method corrects.
>   3.  **Reasoning Length vs. Difficulty:** A correlation study between the length of the thinking trace (i.e., number of tokens) and problem difficulty and types, providing insight into how the model dynamically allocates computational effort.
>
>
> ### 3. **On the Impact of the SSR Buffer Size**
>
> The reviewer raises a valid concern regarding the SSR buffer size and the risk of learning from stale, off-policy data. A very large buffer can indeed degrade performance by increasing the discrepancy between the behavior policy and the current policy. While a full ablation on buffer size was not feasible during the rebuttal period due to computational limitations, we recognize its importance. We commit to including this ablation and a discussion on it in the final version of the paper.
>
> ### 4. **On the Robustness of Hyperparameters ($\alpha$ and Rethinking Fraction)**
>
> We appreciate the feedback on Figure 9 and the general question of hyperparameter robustness.
>
> * **Alpha ($\alpha$) Sensitivity:** The range for $\alpha$ in Figure 9 was selected to demonstrate performance stability across a set of reasonable values. But we realize the current manuscript does not justify this range is reasonable by connecting it to the sharpness of the temperature-annealed softmax distribution. We will refine this in our revision.
> * **Rethinking Hyperparameters:** we will add a brief discussion in the Appendix on the sensitivity to the rethinking fraction and the selection of triggers, providing practical guidance for applying our method to new problem settings.
>
> ### 5. **Clarifications and Minor Points**
>
> * **Figure 2 Training Setting:** Thank you for requesting clarification. This figure illustrates a 72B-Instruct model trained with standard GRPO on publicly available datasets. We will update the caption and text to make this clear. The figure's purpose is to motivate our work by showing that even large models can quickly saturate their performance on publicly available VL reasoning queries that are potentially noisy, and easy-to-learn for larger models.
> * **Minor Weaknesses:** We are grateful for these detailed corrections. We will remove the incorrect bolding of MMMU in Table 1, add the missing subscript for $\pi$ on L134, and add direct cross-references to the Appendix sections as suggested.
>
> Thank you once again for your constructive and valuable feedback.

---

> > ### Comment · Reviewer_J7nB · 2025-08-05
> >
> > Thank you for the response. Adding the planned revisions on analyzing reasoning outputs and SSR buffer size will strengthen the paper.

---

> > > ### Author Response · Authors · 2025-08-05
> > >
> > > Thank you once again for your time and valuable guidance. We have finished the qualitative analysis and is now running experiments with varying buffer sizes. These meaningful additions have helped improve our paper a lot.

---

### Official Review · Reviewer_Zs28 · 2025-07-03

**Clarity:** 3
**Significance:** 3
**Originality:** 2
**Rating:** 5
**Confidence:** 3

**Summary:**

This paper presents VL-Rethinker, a method that leverages reinforcement learning (RL) to enable vision-language (VL) models to exhibit slow-thinking capabilities in VL reasoning tasks. By introducing Selective Sample Replay (SSR) to address the vanishing advantages problem in GRPO and Forced Rethinking to encourage explicit self-reflection, the model achieves state-of-the-art (SOTA) performance on multiple multimodal benchmarks.

**Questions:**

I must note that I am not an expert in this field. The following questions are what I am not clear about after reading the paper.
1. If slow-thinking (even forced reasoning or thinking mechanisms) can enhance VL reasoning capabilities, I'm curious if the training process for those VLMs (e.g. Qwen2.5-VL) already uses a similar approach. If not, why?
2. In Figure 1, VL-Rethinker-72B outperforms OpenAI-o1 on the first three datasets (e.g., EMMA full, MathVerse testmini) but lags behind on the latter two (e.g., MMMU-Pro overall, overall). What factors contribute to this inconsistent performance across different benchmarks?
3. The paper mentions that "Understanding why RL incentivizes reflection differently in multimodal contexts versus math-only settings is an important avenue for future work." Do the authors have any initial hypotheses or speculative explanations for this observed difference?
4. It would be valuable to see the performance of the proposed approach across additional benchmarks, including MMStar[1], M3CoT[2], AI2D[3], ChartQA[4].

[1] Lin Chen, Jinsong Li, Xiaoyi Dong, Pan Zhang, Yuhang Zang, Zehui Chen, Haodong Duan, Jiaqi Wang, Yu Qiao, Dahua Lin, et al. Are we on the right way for evaluating large vision-language models? arXiv preprint arXiv:2403.20330, 2024

[2] Qiguang Chen, Libo Qin, Jin Zhang, Zhi Chen, Xiao Xu, and Wanxiang Che. M3Cot: A novel benchmark for multi-domain multi-step multi-modal chain-of-thought. arXiv preprint arXiv:2405.16473, 2024

[3] Aniruddha Kembhavi, Mike Salvato, Eric Kolve, Minjoon Seo, Hannaneh Hajishirzi, and Ali Farhadi. A diagram is worth a dozen images. In Computer Vision–ECCV 2016: 14th European Conference, Amsterdam, The Netherlands, October 11–14, 2016, Proceedings, Part IV 14, pages 235–251. Springer, 2016

[4] Ahmed Masry, Do Xuan Long, Jia Qing Tan, Shafiq Joty, and Enamul Hoque. Chartqa: A benchmark for question answering about charts with visual and logical reasoning. arXiv preprint arXiv:2203.10244, 2022

**Ethical Concerns:**

["NO or VERY MINOR ethics concerns only"]

**Final Justification:**

The reviewer answered all my review comments and used experiments to prove the effectiveness of their method, so I increased the score accordingly.

**Limitations:**

See in Questions.

**Quality:**

3

**Strengths And Weaknesses:**

# Strengths
- The identification and mitigation of the "Vanishing Advantages" problem in GRPO through SSR is a valuable technical insight, addressing a critical limitation in RL for VLMs and improving training stability.

- Forced Rethinking is a novel and lightweight strategy that effectively promotes deliberate reasoning, as evidenced by the model's ability to self-correct and verify answers, leading to state-of-the-art results on key benchmarks.

# Weaknesses
See in Questions.

---

> ### Author Rebuttal · Authors · 2025-07-29
>
> We thank the reviewer for their thoughtful feedback and insightful questions. We are glad to address them below.
>
> **1. Regarding the inclusion of "slow-thinking" data in the original Qwen-VL training.**
>
> This is an excellent question. Based on our analysis of the base model (Qwen2.5-VL), the model does not intrinsically generate reflective reasoning process (see line 152), which is precisely the motivation for "forced rethinking".
>
> There are several potential reasons for not including "slow-thinking" data in Qwen2.5-VL. First, the release of Qwen2.5-VL is within one month after the release Deepseek-R1, so the training data might not include much slow-thinking data. Besides, Qwen-VL are engineered for comprehensive performance across a wide spectrum of tasks, including many "fast-thinking" perception-focused benchmarks (e.g., OCR, grounding) where slow-thinking is less necessary. Second, creating a well-balanced instruction-tuning dataset that effectively mixes "slow-thinking" data for reasoning with "fast-thinking" data for perception is a significant challenge. In fact, a very recent work, MiMo-VL, highlights this difficulty and proposes a complex four-stage tuning process to empower VLMs with slow-thinking capabilities, underscoring the non-trivial nature of this problem.
>
> **2. On the performance gap between VL-Rethinker and OpenAI-o1 on certain benchmarks.**
>
> The reviewer correctly observes the performance discrepancy on benchmarks like MMMU-Pro and MEGA-Bench. These benchmarks are exceptionally broad, covering a massive range of multidisciplinary knowledge and long-tail, real-world visual-language tasks. The sheer diversity of domains and instructions in these tests requires vast world knowledge that is not well-represented in publicly available academic datasets.
>
> Proprietary models like OpenAI's GPT-4o are trained on datasets of a scale and quality that are orders of magnitude larger than what is available to the public. This extensive, high-quality private data likely provides them with a decisive advantage on these benchmarks.
>
>
> **3. On the speculative reasons why RL incentivizes reflection differently in multimodal contexts.**
>
> Our primary hypothesis, mentioned briefly in the paper (line 152), is that the base VLM generate insufficient reasoning processes that include self-reflection in its native sampling distribution. In more technical terms, during the RL exploration phase, the model's rollouts overwhelmingly consist of direct, concise answers rather than multi-step, reflective thought processes.
>
> This creates an exploration bottleneck. Since the model rarely, if ever, spontaneously produces the "slow-thinking" behaviors we want to reward, the RL algorithm has few opportunities to learn and reinforce these complex reasoning pathways. This contrasts with text-only settings, where the base models have seen slow-thinking data more frequently in pre-training and instruction tuning, and samples these reflective responses during RL more frequently.
>
> **4. Regarding evaluation on additional benchmarks.**
>
> We appreciate the reviewer suggesting these highly relevant benchmarks. Our selection of the seven benchmarks used in the paper was deliberate, aiming to cover a representative set of reasoning skills. For instance, MathVerse, MathVista, MathVision assess quantitative reasoning skills similar to those in **MMStar**. Likewise, MathVerse, EMMA, MMMU involves evaluating diagram and chart understanding capabilities that overlap with **AI2D** and **ChartQA**.
>
> We acknowledge the value of broadening our evaluation and will certainly consider including these benchmarks in future iterations of our work to provide an even more comprehensive assessment.

---

### Official Review · Reviewer_UJQZ · 2025-07-07

**Clarity:** 3
**Significance:** 3
**Originality:** 3
**Rating:** 5
**Confidence:** 4

**Summary:**

This paper introduces a method to facilitate reasoning in vision-language models. Authors propose to add selective sample replay to avoid the issue of diminishing advantages in GRPO, which in turn stagnates RL training, preventing the models from learning more complex reasoning patterns. Additionally, to encourage more reasoning steps, authors also propose to append a rethinking token to encourage longer reasoning rollouts. These two changes allow the model to excel in visual reasoning on a variety of benchmarks, with impressive performance on MathVista and MathVerse in particular.

**Questions:**

- What is the definition 'effective queries' in Figure 2?
- How do you build figure 6? Is this at the end of training? Or average throughout training? I think a better way to show this would be to show a plot with update steps or epochs on the x axis, and either fraction of low-advantage queries or average absolute advantage on the y axis, then show lines for GRPO, GRPO filter, and GRPO SSR. Your claim in the paper is that without SSR, model quickly reaches very low advantage values, but you don't show that anywhere with an experiment. This plot would address this problem.
- How do you get the word cloud in Figure 4? I don't think this figure adds any value to the paper, I think you can just remove it.
- In line 169, you say you incrorporate the additional SFT loss. Is training with this loss happening at the same time as the RL stage, or is it separate?

### Typos and nitpicks
- The text in Figure 3 is too small
- Line 132: exhibited non-zero : I think you missed 'advantage' here.
- Line 134: Breplay - replay should be in subscript like it is in line 131.

**Ethical Concerns:**

["NO or VERY MINOR ethics concerns only"]

**Limitations:**

See comments above

**Quality:**

3

**Strengths And Weaknesses:**

**Strengths**
 - I like the intuitive simplicity of the proposed ideas
- Thorough experiments including large-scale models (72B), and impressive performance across a variety of benchmarks

**Weaknesses**
- The authors claim that without SSR, the model quickly reaches very low advantage values, but this is not confirmed anywhere in the experiments.
- Using methods akin to prioritized experience replay can lead to known issues in RL, such as oversampling outliers or outdated samples from earlier in the training. These issues may become more prononunced as RL stage is scaled further.

---

> ### Author Rebuttal · Authors · 2025-07-29
>
> We sincerely thank the reviewer for their patience of reading the paper and their constructive feedback. The suggestions have greatly helped us improve the clarity and impact of our paper. Below, we address each of the questions raised.
>
> **1. Definition of 'effective queries' in Figure 2.**
>
> We appreciate the reviewer's request for clarification. In the context of Figure 2, "effective queries" refers to queries for which the corresponding training batch contains at least one non-zero advantage value, thus effectively contributing to the gradient update.
>
> More specifically, the GRPO method computes advantages within each query group. A key scenario arises when all responses within a single query group receive an identical reward (e.g., all rewards are 0 or all are 1). In this case, the calculated advantages for all samples in that group become exactly zero. Consequently, these samples do not provide a learning signal (i.e., they contribute zero gradient) during the policy update.
>
> Currently the direct definition of "effective queries" is missing, which can lead to readers' confusions. Since the notion of "effective queries" is indeed equivalent to the number of "effective training samples" (query-responses with non-zero advantages), in our revision, we will use the more unambiguous term "effective training samples" to avoid any confusions.
>
> **2. Clarification and Improvement of Figure 6.**
>
> We thank the reviewer for this suggestion. The current Figure 6 was generated via a post-hoc analysis: we used a static set of training rollouts and applied three distinct filtering methods (no filtering, zero-advantage filtering, and our SSR) to this fixed dataset to plot the resulting advantage histograms.
>
> We agree that a dynamic plot tracking advantage metrics over the course of training would be far more compelling and would directly substantiate our claim that advantage values diminish rapidly without SSR. In the revised manuscript, we will replace the current figure with a new plot following the reviewer's suggestions.
>
> **3. Rationale for Figure 4.**
>
> The word cloud in Figure 4 was generated by sampling responses from our trained VL-Rethinker on the validation set and visualizing the most frequent words. We agree with the reviewer's assessment that this figure is primarily illustrative and does not contribute significant technical value to the paper's core claims. To streamline the paper and focus on the most impactful results, we will remove this figure in our revision.
>
> **4. The Role of the Additional SFT Loss.**
> The SFT loss is an auxiliary loss that is applied concurrently with the RL objective during the RL training phase. Its purpose is to directly incentivizes the policy to adopt the "forced rethinking" responses. While the model can learn this behavior through the RL objective alone, we found that the learning process is significantly slower. This is because the "wait" token initially has a low probability in the policy's native distribution, causing the policy ratio in the PPO objective to be frequently clipped. The auxiliary SFT loss thus provide more direct gradients that accelerates the adoption of this rethinking token.
>
> **5. Typos and Nitpicks**
>
> We are grateful to the reviewer for their careful reading and for catching these issues. We will correct them in the revised version.

---

> ### Comment · Area_Chair_4QBn · 2025-08-08
> **final score**
>
> Dear Reviewer
>
> Could you please check the rebuttal and update the final score ?
>
> Best,
> AC

---

### Author Response · Authors · 2025-08-01
**Summary of Reviews and Author Rebuttal**

Dear Reviewers, AC, and SAC,

We would like to express our sincere gratitude for your comprehensive and insightful feedback on our paper. We are very encouraged that the reviewers recognized the novelty of our work, the strength of our empirical results, and the significance of the problem we are addressing. **We are pleased to find there is a consensus on the following positive points:**

**Novelty and Contribution:** The reviewers unanimously found our proposed techniques, Selective Sample Replay (SSR) and Forced Rethinking, to be novel and effective.
- Reviewer UJQZ recognized "the intuitive simplicity of the proposed ideas"
- Reviewer Zs28 highlighted that "The identification and mitigation of the "Vanishing Advantages" problem in GRPO through SSR" is "addressing a critical limitation in RL for VLMs and improving training stability."
- Reviewer EyWX praised that the proposed training recipe "demonstrating scalability and domain generality without additional supervision"
- Reviewer J7nB also noted the "valuable technical insight" of the proposed methods and praised that the paper "addresses the important problem of reasoning for multimodal tasks, where previously, multimodal reasoning capabilities were similar to non-reasoning models."


**Strong Empirical Effectiveness:** The reviewers acknowledged the strength and impressiveness of our results across numerous benchmarks.
- Reviewer UJQZ noted the "impressive performance across a variety of benchmarks".
- Reviewer Zs28 mentioned "the model achieves state-of-the-art (SOTA) performance on multiple multimodal benchmarks."
- Reviewer J7nB praised the "strong empirical performance" ,
- Reviewer EyWX highlighted that our model "achieves state-of-the-art open-source performance" and "scales robustly across model sizes".


**Thorough Experimentation and Analysis:** We are glad the reviewers found our experimental evaluation to be rigorous.
- Reviewer UJQZ commented on the "Thorough experiments" ,
- Reviewer J7nB stated that "The paper has comprehensive analyses ablating the components of VL-Rethinker".

**Reproducibility and Community Value:** The reviewers recognized the value of fostering future research in the community.
- Reviewer J7nB noted that "The high-quality dataset of 38,870 queries is also valuable for future work to build off of" and the paper "provides sufficient implementation details in Appendix B."
- Reviewer EyWx acknowledged that this work is "offering a reproducible and scalable framework"

We deeply appreciate these positive comments. **In addition to this encouragement, the reviewers provided constructive suggestions for further analysis, clarifications of technical details, and improvements to the paper's presentation. We have carefully considered all feedback and have addressed each point in our individual responses.**

Based on this valuable guidance, **we will incorporate the following revisions into our manuscript. We have already completed most of the requested experiments and analyses.**
- Improve Clarity of the Method:
  - The role of addtional SFT loss for forced Rethinking (Reviewer UJQZ, J7nB). We will make clear that this loss aims to accelerate the adoption of rethinking tokens.
- Enhance Figure Presentations
  - Replace with "effective training samples" in Figure 2 (Reviewer UJQZ)
  - Remove Figure 4 (Reviewer UJQZ)
  - Use dynamics-tracking plot for Figure 6 (Reviewer UJQZ)
- New Results and Analysis
  - Additional Evaluations on ChartQA, MMStar, AI2D (Reviewer Zs28)
  - Analysis of Reasoning outputs:
    - tokens and computation efficiency across difficulties (finished) and problem types (Reviewer J7nB)
    - success and failure analysis of rethinking (finished, Reviewer EyWX)
  - Robustness to hyperparameters:
    - make explicit the connection between selected range of alpha with the sharpness of temperature-annealed softmax distribution (Reviewer J7nB)
    - new analysis and discussion on the impact of replay buffer size (Reviewer J7nB)
  - results to support modality-agnostic property of SSR (finished, Reviewer EyWX)
- Expanded Discussion: we will add a dedicated section on the outcome supervision setting, downsides of SSR (per Reviewer EyWX) and speculations about gap in MMMU and distinctions of RL self-reflections (per Reviewer Zs28)

We believe these revisions will address the reviewers' valuable feedback and significantly strengthen the final version of our paper. Thank you once again for your time and constructive guidance.

---

### Note · Authors · 2025-08-13

We sincerely thank all reviewers and the Area Chair for their time and valuable contributions to improving our manuscript.

**We are encouraged that the reviewers reached a consensus on several key strengths of our work**, including its novelty and contribution, strong empirical effectiveness, thorough experimental validation, and value to the community.

**The reviewers also raised insightful questions and provided constructive suggestions for improvement.** The primary points centered on:
- clarifications on the additional SFT loss for forced rethinking
- deeper analysis into the rethinking behaviors and reasoning outputs
- further ablation studies to better understand hyperparameter choices of SSR buffer size
- empirical support for the modality-agnostic claim of SSR.

**We have addressed these points individually in our rebuttals.** We provided the requested new analyses and experimental results, and clarified all ambiguous points.

Based on these constructive feedback, **we also outlined the actionable revision plans**, as detail in the rebuttal summary below.

Thank you once again for the valuable time and guidance.


Authors of Submission-10204

---

### Decision · Program_Chairs · 2025-09-17

**Decision:**

Accept (spotlight)

**Comment:**

All reviewers agree on the paper's acceptance.

- The proposed Selective Sample Replay (SSR) and Forced Rethinking, are novel and effective.
- Strong empirical effectiveness.
- Thorough experiments.